# Arrhythmic Mitral Valve Prolapse in the Young: A Rare but Concerning Entity

**DOI:** 10.3390/diagnostics12071519

**Published:** 2022-06-22

**Authors:** Nicolò Martini, Alberto Cipriani, Bortolo Martini, Barbara Bauce, Martina Perazzolo Marra, Sabino Iliceto, Domenico Corrado

**Affiliations:** 1Department of Cardio-Thoraco-Vascular Sciences and Public Health, University of Padua, Via Giustiniani, 2, 35128 Padua, Italy; nicolo.martini.2@studenti.unipd.it (N.M.); barbara.bauce@unipd.it (B.B.); martina.perazzolomarra@unipd.it (M.P.M.); sabino.iliceto@unipd.it (S.I.); domenico.corrado@unipd.it (D.C.); 2Cardiac Unit, Alto Vicentino Hospital, 36014 Santorso, Italy; bortolo.martini@gmail.com

**Keywords:** mitral valve prolapse, ventricular arrhythmias, sudden cardiac death, pediatric age, cardiac imaging techniques

## Abstract

Arrhythmic mitral valve prolapse (MVP) is an increasingly recognized clinical entity, characterized by the association of myxomatous mitral valve, ventricular arrhythmias (VAs) and sudden cardiac death (SCD). Prevalence of MVP is reported ranging between 2% and 5% of the general population, and risk of SCD is estimated approximately 0.3% per year. Diagnosis of MVP and the occurrence of fatal events involve generally adults aged 30 to 50 years, whereas in younger and even pediatric individuals has rarely been described. Herein, we report two clinical cases of malignant MVP in young patients, with the aim to point out the clinical features and the challenge of clinical management and risk stratification.

**Case 1**. A 11-year-old boy was referred to our hospital because of frequent (>500/24 h) premature ventricular contractions (PVCs) detected during sport pre-participation screening. Family history was examined, revealing a grandfather with hypokinetic cardiomyopathy, but no SCD cases. Physical examination detected a mid-systolic click and a mild systolic regurgitation murmur. Electrocardiogram (ECG) was unremarkable, except for non-specific repolarization abnormalities on inferior leads (Figure 1A). A 24-h Holter ECG monitoring and treadmill stress test allowed a more in-depth investigation of PVCs, showing a predominant right bundle branch block morphology, persistence during effort, but no repetitive or complex patterns (Figure 1B). Two-dimensional echocardiogram (2D-echo) showed a bileaflet MVP with mild regurgitation, mitral annular disjunction (MAD) and a mild dilatation of the left ventricle (LV) (Figure 1D,E). Given the arrhythmias and echocardiographic findings, cardiac magnetic resonance (CMR) was prompted, and revealed thickened, myxomatous and prolapsing mitral valve leaflets, a 10 mm MAD, curling of the posterior mitral ring and a LV mild dilatation and systolic dysfunction. No late gadolinium enhancement (LGE) was documented (Figure 1C–F). Given the family history for cardiomyopathy, genetic testing was performed, but no genetic variants were identified. A low dosage beta blocker therapy (metoprolol 25 mg bid) was then started. The patient remained symptoms free with an overall reduction of the PVCs’ burden. The patient remained asymptomatic during a 2-year follow-up, and a significant reduction of PVC burden was observed.

**Case 2**. A girl aged 14 years old with unremarkable medical and family history, came to our attention for palpitations. ECG showed inferior T-wave inversion and lateral flattened T-wave (Figure 2A). Holter ECG pointed out frequent PVCs and also non-sustained ventricular tachycardia (NSVT) episodes. 2D-echo revealed a bileaflet mitral valve prolapse with mild regurgitation, a broad (9 mm) MAD and curling of the posterior mitral ring (Figure 2B–E). Biventricular kinetic abnormalities (hypokinesia of the left ventricular posterior and lateral walls, hypokinesia of the right ventricular lateral wall and of the sub-tricuspid region) and mild reduction of the ejection fraction (EF) were also documented. CMR emphasized the mitral valve apparatus and LV abnormalities, also detecting intramural non ischemic LGE in the lateral-basal wall of LV (Figure 2C,D). In the end of the diagnostic work-up, genetic testing was also performed and no genetic variants were detected. Low dosage of metoprolol (25 mg, bid) was started with partial relieve of symptoms and arrhythmias during a 3-year follow up”. 

Management and risk stratification of MVP are known to be challenging [1]. The natural history of this condition is heterogeneous, ranging from a benign clinical course with no or minor symptoms, to more severe conditions such as flail leaflets mitral regurgitation, infective endocarditis, acute stroke and SCD. Among all, arrhythmic MVP is a peculiar entity with typical features: myxomatous leaflets, MAD, hypermobility of the valve and systolic curling of the posterior mitral annulus with a paradoxical increase in the systolic annulus diameter, all supposed to contribute to an abnormal mechanical stress of the myocardium and development of replacement fibrosis [2,3,4]. Given the high prevalence of the echo-defined MVP in the general population, the identification of the arrhythmic MVP is mandatory. To date, however, a definite risk stratification strategy for SCD lacks, so that the management of MVP patients with VAs, both adults and pediatrics, is still empiric and relies on single center’s experience. MVP in pediatrics and young individuals is presumed to be a rare entity, despite more precise epidemiological data are lacking. In a SCD autopsy series reporting MVP, the mean age of victims was 30 years [4]. Similarly, in other arrhythmic MVP cohorts, adult patients were much more prevalent. This may reflect a time-dependent pathophysiology of this condition: electrical instability may occur long after the myocardial remodeling due to the abnormal mitral valve morphology and function [5]. According to this view, the two cases reported appear to be different stages of the natural history of malignant MVP, with the girl succeeding the boy, due the detection of frequent NSVTs and LV LGE. Clinical management of patients such as these is problematic, dealing with VAs control and SCD risk stratification in a less-recognized clinical entity such as arrhythmic MVP. Sport disqualification, antiarrhythmic therapy, ICD implantation are all measures that need to be addressed. Although SCD in MVP patients usually occurs while at rest or during sleep [5], competitive sport eligibility in young patients with arrhythmic MVP, such as those here reported, is questionable and needs accurate SCD risk stratification [6]. Moreover, the risk of mitral valve disease progression due to sport activity is unknown. Antiarrhythmic drugs such as beta blockers, can be safely used in pediatrics to control VAs [3], particularly in presence of symptoms and repetitive or complex VAs. Implantable cardioverter defibrillator (ICD is still proposed as a secondary prevention treatment in MVP, even if some individuals with multiple arrhythmic markers, such as LGE, MAD, NSVT, may benefit from primary prevention [3]. It must be considered, however, that ICD implantation during childhood and youth carries a higher risk of severe complications (lead failure, infective endocarditis, inappropriate shocks during sport activities), thus its implementation requires caution and further evidence. 

Herein we report two rare cases of arrhythmic MVP in pediatric patients. Clinical and anatomical features appear to be similar to those reported in adulthood. Risk stratification and clinical management are, however, even more challenging given the younger age and the lack of data about prevalence, evolution and prognosis. Pediatricians and sport medical doctors should be aware of this rare clinical condition to provide early diagnosis, VAs treatment and manage follow-up of both VAs and mitral valve abnormality.

## Figures and Tables

**Figure 1 diagnostics-12-01519-f001:**
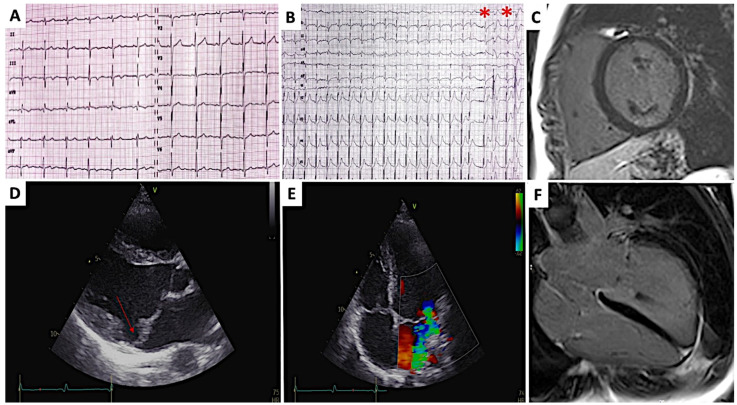
Male 11-year-old patient. (**A**). 12-leads ECG showing sinus rhythm, non-specific repolarization abnormalities on inferior leads (II, III, aVF). (**B**). 12-leads ECG recorded during the recovery phase of the treadmill test revealing premature ventricular contractions of right bundle branch block morphology (red asterisks). (**D**). 2D-echocardiogram in parasternal long axis view showing redundant and prolapsing mitral valve leaflets, MAD (red arrow) and mild dilatation of the left ventricle. (**E**). Apical 4-chamber view showing mitral valve regurgitation mitral and tricuspid valve prolapses. (**C**,**F**). Cardiac magnetic resonance imaging. Late post contrast long and short axis images showing no sign of myocardial hyperenhancement.

**Figure 2 diagnostics-12-01519-f002:**
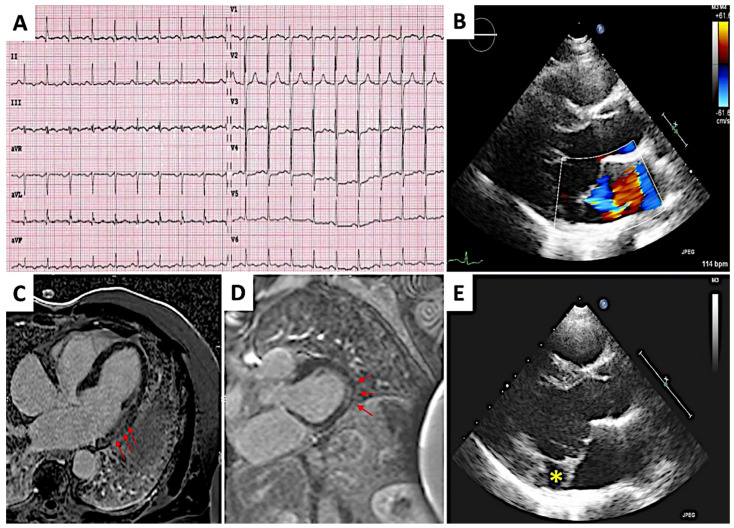
Female 14-year-old patient. (**A**). 12-leads ECG showing repolarization abnormalities on inferior and lateral leads. (**B**,**E**). 2D-echocardiogram with parasternal long axis view and Doppler signal reporting redundant and prolapsing mitral valve leaflets with mild regurgitation, MAD (yellow asterisk) and mild dilatation of the LV. (**C**,**D**). Cardiac magnetic resonance imaging showing intramural late gadolinium enhancement in the lateral basal wall of the left ventricle (red arrows).

## Data Availability

The data presented in this study are available on request from the corresponding author.

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
