# Peer review of "Arrhythmic Mitral Valve Prolapse in the Young: A Rare but Concerning Entity"

_diagnostics, 2022, doi:10.3390/diagnostics12071519_

Round 1

Reviewer 1 Report

Martini et al presented two cases of arrhythmic mitral valve prolapse. It is a condition with increasing interest and various prognosis lacking relevant information regarding risk stratification especially in pediatric patients. The illustration of the manuscript is of high standard. However, this paper lacks relevant information regarding the diagnostic methods and the follow-up as well.

Major:

1.       In case two the presence of RV hypokinesia raises the question if this clinical presentation is simply biventricular valve prolapse and annular disjunction or coexistence of a cardiomyopathy and valvular disease. In my opinion in a case report it would be fundamental to provide genetic information (patient 1 had positive family history for cardiomyopathy).

2.       In the discussion section the authors suggest that the two cases reported appear to be different stages of the natural history of malignant MVP. In my opinion it should be stated only if the authors also provide genetic information.

3.       You mention follow-up in both cases. Please define the length of the follow-up. How do you manage the follow-up? How often do you perform echocardiography/CMR?

4.       Based on literature data additional imaging techniques including speckle tracking echocardiography or T1 mapping/ECV could add relevant information to the risk stratification. Could you provide these values in the two cases?

5.       In the conclusion the author highlight complications such as infective endocarditis and acute stroke. How common are these complications – especially in pediatrics? Do we have any data regarding indication of infective endocarditis or anticoagulation in MAD patients?

In the conclusion the authors stated that „Pediatricians and sport medical doctors need to be aware of this rare clinical condition to provide early diagnosis, VAs treatment and prevention of major complications.” What would prevention of major complication refer to in the two reported cases (except adding beta blocker in both cases)?

Minor:

6.       Could you define „frequent” in the term of PVCs. Did frequent PVCs cause difficulties in ECG synchronization during CMR examination? Did you experience arrhythmia related artifacts on cine or LGE images?

7.       Please define „broad MAD”, add the extent size of MAD.

Author Response

Martini et al presented two cases of arrhythmic mitral valve prolapse. It is a condition with increasing interest and various prognosis lacking relevant information regarding risk stratification especially in pediatric patients. The illustration of the manuscript is of high standard. However, this paper lacks relevant information regarding the diagnostic methods and the follow-up as well.

R: We thank the reviewer for the positive comments.

Major:

  1. In case two the presence of RV hypokinesia raises the question if this clinical presentation is simply biventricular valve prolapse and annular disjunction or coexistence of a cardiomyopathy and valvular disease. In my opinion in a case report it would be fundamental to provide genetic information (patient 1 had positive family history for cardiomyopathy).

R: We agree with the Reviewer’s comments. Genetic testing was performed in both patients; however, no genetic variants were identified, thus, a gene-based diagnosis or diagnostic hypothesis of cardiomyopathy could not be provided. Accordingly, we have rephrased the manuscript as follows: page 1, line 36 “Given the family history for cardiomyopathy, genetic testing was performed, but no genetic variants were identified” and page 2, line 58 “In the end of the diagnostic work-up, genetic testing was also performed and no genetic variants were detected.”.

  1. In the discussion section the authors suggest that the two cases reported appear to be different stages of the natural history of malignant MVP. In my opinion it should be stated only if the authors also provide genetic information.

R: Genetic testing was performed in both patients. but was negative.

  1. You mention follow-up in both cases. Please define the length of the follow-up. How do you manage the follow-up? How often do you perform echocardiography/CMR?

R: The length of the follow up was two years for the first case and three years for the second case. We usually manage follow up on outpatient basis with a dedicated path for patients affected by cardiomyopathies or arrhythmic syndromes. Outpatient visits, along with electrocardiogram and echocardiogram, usually occur once a year, while CMR is usually performed every 3-5 years, or before, in case of newly detected echocardiographic or electrocardiographic findings. Accordingly, we have rephrased the manuscript as follows: page 1, line 38 “The patient remained asymptomatic during a 2-year follow-up, and a significant reduction of PVC burden was observed” and page 2, line 59 “Low dosage of metoprolol (25 mg, bid) was started with partial relieve of symptoms and arrhythmias during a 3-year follow up

  1. Based on literature data additional imaging techniques including speckle tracking echocardiography or T1 mapping/ECV could add relevant information to the risk stratification. Could you provide these values in the two cases?

R: As the reviewer correctly said, additional imaging techniques have been proposed to implement the diagnostic power of echocardiography and CMR in MVP. Such improvements could be useful to detect early signs of fibrosis (T1 mapping) or contraction abnormalities of the basal left ventricular myocardium (speckle tracking) in patients with MVP (Guglielmo M, et al. T1 mapping and cardiac magnetic resonance feature tracking in mitral valve prolapse. European Radiology. 2020;31(2):1100-1109. Pavon A, et al. Myocardial extracellular volume by T1 mapping: a new marker of arrhythmia in mitral valve prolapse. Journal of Cardiovascular Magnetic Resonance. 2021;23(1). Palmer C, et al. Contractile Differences Detected by Speckle Tracking Echocardiography in Pediatric Patients with Mitral Valve Prolapse. Pediatric Cardiology. 2021;42(8):1706-1712). In our cases, mapping techniques were not performed because unfortunately not available in our center at that time. Speckle tracking evaluation was not included in the exam, given that wall motion abnormalities were overt.

  1. In the conclusion the author highlights complications such as infective endocarditis and acute stroke. How common are these complications – especially in pediatrics? Do we have any data regarding indication of infective endocarditis or anticoagulation in MAD patients?

R: This a crucial point. However, given the rarity, and the probable under recognition of arrhythmic MVP in pediatrics and young patients, data about prevalence of MVP-related infective endocarditis, acute stroke or anticoagulant therapies are lacking. Further evidences are required to better define a causal relationship as well as provide indications for specific therapy.

In the conclusion the authors stated that "Pediatricians and sport medical doctors need to be aware of this rare clinical condition to provide early diagnosis, VAs treatment and prevention of major complications.” What would prevention of major complication refer to in the two reported cases (except adding beta blocker in both cases)?

R: We agree with the Reviewer that this was a too strong and aspecific statement. We have rephrased as follows.

Page 4, line 105. “Pediatricians and sport medical doctors should be aware of this rare clinical condition to provide early diagnosis, VAs treatment and manage follow-up of both VAs and mitral valve abnormality”.

Minor:

  1. Could you define “frequent” in the term of PVCs. Did frequent PVCs cause difficulties in ECG synchronization during CMR examination? Did you experience arrhythmia related artifacts on cine or LGE images?

R: when we use the term “frequent”, we intend a total of PVCs >500/24 h. As correctly pointed out by the Reviewer, the burden of PVCs usually may affect CMR examination due to artifacts and poor cardio-synchronization. Fortunately, we didn’t have so many difficulties regarding artifacts or ECG synchronization, in one case because the PVC burden was low during CMR, in the other because a low dosage of metoprolol was administered before the diagnostic exam. Accordingly, the manuscript was modified as follows: page 1, line 22 “A 11-year-old boy was referred to our hospital because of frequent (>500/24 hours) premature ventricular contractions (PVCs)”.

  1. Please define “broad MAD”, add the extent size of MAD.

R: as suggested, we have added the dimension of the MAD as follows: page 2, line 53 “mild regurgitation, a broad (11 mm) MAD and curling of the posterior mitral ring”.

Reviewer 2 Report

This case report by Martini et al presented two clinical cases of malignant arrhythmic mitral valve prolapse in young patients. Generally, this unusual observation could be valuable for other researchers in this field. Some comments were suggested as follows.

1. In my experience, for the case report manuscript that includes details and images relating to an individual person, written informed consent for the publication of these details must be obtained from that person or the legal guardian, especially in the case of young patients under 18. However, I don’t find any relevant information in this manuscript.

2. Line 14: “and risk of SCD is estimated approximately 0.3% per year [1,3].” Please avoid citing references in the Abstract.

3. The quality of Figure 2D is poor. I recommend replacing it with high-quality one.

Author Response

This case report by Martini et al presented two clinical cases of malignant arrhythmic mitral valve prolapse in young patients. Generally, this unusual observation could be valuable for other researchers in this field.

R: we thank the reviewer for the positive comments.

Some comments were suggested as follows.

  1. In my experience, for the case report manuscript that includes details and images relating to an individual person, written informed consent for the publication of these details must be obtained from that person or the legal guardian, especially in the case of young patients under 18. However, I don’t find any relevant information in this manuscript.

R: We agree with the Reviewer. Written informed consent were obtained by the family for all the diagnostic processes involving the two young patients and for the publication of clinical and imaging details. Accordingly, we have added this information in the manuscript.

  1. Line 14: “and risk of SCD is estimated approximately 0.3% per year [1,3].” Please avoid citing references in the Abstract.

R: We thank the Reviewer for the suggestion.

  1. The quality of Figure 2D is poor. I recommend replacing it with high-quality one.

R: Unfortunately, this is the highest quality of the image we could provide. We tried to modify some parameters of the image setting to enhance resolution.

Round 2

Reviewer 1 Report

Thank you for your comments and corrections.